# Revisiting Minamata disease through computational phenotypic similarity analysis

Edoardo Marchi[1,2], Paolo Boldi[2,3], Elena Casiraghi[3], Stefano Zapperi[2,4,5], Caterina A. M. La Porta[1,2,6]*

1 Department of Environmental Science and Policy, Università degli Studi di Milano, Milan, Italy, 2 Center for Complexity and Biosystems, University of Milan, Milan, Italy, 3 Department of Computer Science, Università degli Studi di Milano, Milan, Italy, 4 Department of Physics, Università degli Studi di Milano, Milan, Italy, 5 CNR - Consiglio Nazionale delle Ricerche, Istituto di Chimica della Materia Condensata e di Tecnologie per l'Energia, Milan, Italy, 6 Fondazione IRCCS Ca' Granda, Ospedale Maggiore Policlinico di Milano, Milan, Italy

* caterina.laporta@unimi.it

## Abstract

Minamata disease, a severe neurological disorder caused by methylmercury exposure in 1950s Japan, is historically recognized for its profound impact on environmental health awareness. However, its phenotypic complexity and potential overlap with other neurological disorders have not been systematically assessed in a modern computational framework. In this study, we adopt a network approach to reinterpret Minamata disease within a broader disease similarity landscape. We mapped clinical symptoms from an extensive epidemiological survey of 269 Minamata patients to standardized Human Phenotype Ontology (HPO) terms, constructing a comprehensive phenotypic profile. Using network-based and computational similarity measures—Jaccard Index, ontology-informed metrics (Resnik and GraphIC), and information retrieval techniques (TF-IDF with query expansion), we compared this profile to over 12,000 diseases. Our results consistently identified strong phenotypic ties between Minamata disease and several movement and neurodegenerative disorders, including cyanide-induced parkinsonism and progressive supranuclear palsy. A weighted rank aggregation across methods revealed a robust consensus network of diseases with overlapping symptomatology, underscoring the systemic nature of these complex neurological disorders. Our study highlights the utility of integrating historical epidemiological data with contemporary network tools to reveal novel associations between environmental exposures and systemic pathophysiological responses. Our findings provide a blueprint for exploring environmentally triggered disease mechanisms and their broader implications for network-based understanding of human disease.

**Data availability statement:** Code and data to reproduce the results in this paper are available at https://github.com/marchiedoardo/Minamata_KG.

**Funding:** CAMLP and EC acknowledge funding from FAIR - Future Artificial Intelligence Research: Adaptive AI methods for Digital Health, funded by the NextGenerationEU program within the PNRR-PE-AI scheme, grant number PNRR_BAC24GVALE_01 PE_0000013, CUP D53C2200238000. EC acknowledges support from the National Plan for NRRP Complementary Investments (PNC) in the call for the funding of research initiatives for technologies and innovative trajectories in the health—project n. PNC0000003—AdvaNced Technologies for Human-centred Medicine (project acronym: ANTHEM). Computational resources at CINECA for this work have been funded by UNITECH INDACO, which is an HPC project at the University of Milan.

**Competing interests:** The authors have declared that no competing interests exist.

## Introduction

Minamata disease is a neurological disorder caused by severe mercury poisoning. It was first identified in the 1950s in Minamata city in Japan, when residents began experiencing symptoms like numbness in the limbs, muscle weakness, and damage to hearing and speech. The disease was traced back to the consumption of fish and shellfish contaminated by methylmercury, a highly toxic compound released into Minamata Bay by the Chisso Corporation's chemical factory. Methylmercury bioaccumulated in the marine food chain, resulting in widespread human exposure [1,2]. Minamata disease had devastating effects on the local community, leading to thousands of deaths and lifelong disabilities. The tragedy of Minamata disease also brought global attention to the dangers of industrial pollution and the importance of environmental regulations and pointed out the toxicity due to heavy metals [3].

Since what happened in Minamata, a series of studies on the impact of heavy metals on humans has begun, showing in particular the highly toxic effect of organic mercury [4,5]. Minamata was an environmental disaster, but even today heavy metals are important pollutants. In fact, they are naturally present in rocks, soil and water and can also be released through natural processes like weathering and volcanic eruptions. Moreover, there are also anthropogenic sources including industrial processes, mining, waste disposal, and agricultural runoff. In a recent paper, our group analyzed concentrations of heavy metals in surface waters and groundwater monitored by the Regional Environment Protection Agency of Lombardy from 2017 to 2020 [6]. The spatio-temporal analysis of heavy metals in the surface waters and groundwater revealed that Lombardy, one of the richest and populous region of Europe, could reduce pollution from these substances over years thanks to the introduction of more restrictive laws [6]. Pollution and its impact on human health is a point that will be increasingly elucidated in the coming years. The most difficult challenge nowadays is to show if there is a relation between human health and polluting factors, including heavy metals, in concentrations not as high as in Minamata, but constant over time and present in the same location together with other pollutants.

The main goal of the present paper is to find diseases with symptoms similar to those found in Minamata's patients in order to single out mechanisms or environmental triggers involved in the development of the pathology. Considering the critical role of heavy metals as pollutants, this comparison could help clinicians in devising more focused treatment strategies. To this end we leverage the HPO database [7], which includes over 18,000 terms describing various abnormalities and features found in human diseases. It also contains over 156,000 annotations linking these terms to hereditary diseases, including both rare and common conditions. HPO does not include, therefore, specific diseases, but rather defines and describes phenotypic abnormalities that are common to various diseases. Previous studies have used similarity of HPO to evaluate disease distance, such as the Rare Disease map (RDmap) [8], which calculates and visualizes distances for thousands of rare diseases. RDmap considers only diseases that are already represented in HPO and is primarily focused on interactive exploration. In contrast, our work uses Minamata disease as a case

study of a disorder that is not natively represented in HPO or OMIM and whose clinical description originates from historical epidemiological surveys. Our study does not aim to introduce a novel similarity algorithm, but to demonstrate a reproducible workflow for translating historical clinical descriptions into ontological form and contextualizing them within the modern landscape of HPO-annotated diseases. Minamata disease serves as the ideal case study because it represents a prototypical environmental neurotoxic disorder with a well-documented clinical picture, yet it is absent from modern disease ontologies and databases. The resulting similarity rankings are intended as hypothesis-generating, not as definitive mechanistic proofs and must be tested in future experimental and epidemiological studies.

## Materials and methods

### Preparation of diseases dataset and data-processing

In this study, we utilized the HPO annotations for diseases file (`phenotype.hpoa`, v2024-08-13), which provides disease–phenotype associations along with additional metadata, including phenotype frequency, publication references, biocuration details, and evidence codes. The database includes 12,630 annotated diseases from three different ontologies: OMIM [9] (8,319 diseases), OrphaNet [10] (4,256 diseases) and DECIPHER [11] (47 diseases), with a total of 268,776 associated terms from the Human Phenotype Ontology (HPO). However, not all HPO terms represent phenotypic abnormalities—for instance, some refer to modes of inheritance or past medical history. Since our focus was on symptoms, we filtered the database to retain only HPO terms related to phenotypic abnormalities. The filtered database is composed of 12,622 distinct diseases and 251,841 HPO annotations. To ensure consistency and improve efficiency of the analysis, the diseases were mapped from their original IDs (OMIM, Orphanet or DECIPHER) to their corresponding Mondo IDs using the `mondo.obo` file [12] (v2025-02-04). The Mondo (Mondo Disease Ontology) provides a unified, curated ontology that integrates multiple disease resources to enable consistent disease annotation and comparison across datasets. Among all disease-symptom annotations, 198,099 include information about symptom frequency. However, this information is presented in heterogeneous formats, requiring additional preprocessing to standardize it. In particular, frequency can be reported as a fraction (7/11 for example, indicating that among 11 patients with the disease, 7 experienced the symptom), as a percentage (for instance 30%, in this case we do not know the total number of patients with the symptom) or as an HPO term that indicates a range of frequencies (for instance 'HP:0040283', which means *Occasional* occurrence, defined as affecting 5–29% of patients; again, the exact patient count is unspecified in this case). First of all, symptoms without a reported frequency were associated an artificial frequency of 0.5; this choice reflects the assumption of a uniform distribution over the possible frequency range $f \in [0, 1]$, which is a reasonable prior in the absence of more specific information. The value 0.5 corresponds to the expected value of such a distribution. To assess robustness of results, we repeated all analyses with extreme choices 0.01 ("very rare" as for HPO frequency classes) and 1.0 ("obligate"). Another challenge involved handling duplicate disease–symptom pairs in the database, which often occurred when the same disease was annotated with overlapping information from multiple ontologies. To address this, we constructed a dictionary mapping each disease to a set of unique HPO terms, each associated with a single frequency. When a disease–symptom pair appeared multiple times with different reported frequencies, we resolved the conflict by applying the following rules:

- If both frequencies were given as fractions (e.g., 7/11 and 3/5), we computed a weighted average, using the denominators as weights to reflect the original sample sizes.
- If one or both frequencies were reported as ranges (e.g., via an HPO term like HP:0040283), we identified the smallest interval compatible with both frequencies and assigned the midpoint of this interval as the new frequency.
- In all other cases (e.g., percentages without sample sizes or incompatible formats), we used a simple arithmetic mean.

## Minamata disease symptoms dataset

Data about the Minamata Disease symptoms is available in [13], resulting from an extensive survey conducted in the early 1970s. It is in the form of a list of symptoms along with the number of affected patients, covering a total of 3,563 individuals from three regions with respectively high, moderate and low exposure to organic mercury: Minamata, Goshoura, and Ariake. The patients are further classified in three categories: 1) diagnosed with Minamata Disease, 2) suspected Minamata Disease and 3) deferred diagnosis. For this study, this dataset has been digitized and converted into a Pandas dataframe [14], keeping only the information concerning the 269 patients from the Minamata Area who were diagnosed with Minamata Disease. Each one of the symptoms was mapped to the HPO terms [7] that were considered the closest to the original definition. The 'Mental Retardation' symptom, present in the original table (see Table 10 at page 115 in [13]), has a very high associated frequency (89.6%), which contradicted the corresponding value reported in another table from the same source (see Table 15 on page 124 in [13]). To resolve this inconsistency, we adopted the frequency from the latter table, which is 0.37%.

## Similarity measures

The similarity between Minamata disease and the diseases in our database, characterised by their list of associated symptoms and frequency, was computed using several different techniques. The simplest idea is to compute a disease-disease similarity based on the number of shared symptoms, for example, the Jaccard index. If we consider two diseases $A = \{x_1^A, ..., x_N^A\}$ and $B = \{x_1^B, ..., x_M^B\}$, where $x_i^A$ and $x_j^B$ are respectively the symptoms of $A$ and $B$, the Jaccard index of the two sets of symptoms is

$$sim_{Jaccard}(A, B) = J(A, B) = \frac{|A \cap B|}{|A \cup B|}. \tag{1}$$

While this measure is straightforward to implement and computationally efficient, it does not account for either the frequency of symptoms or the hierarchical structure of HPO terms.

The Human Phenotype Ontology can be seen as a directed acyclic graph (DAG), where the nodes are the phenotypic abnormalities and the edges represent 'is a' relationships (for instance '*Microcephaly*' is a '*Abnormality of head size*'). Notably, an HPO term can have multiple parent terms, reflecting that phenotypes can belong to more than one broader category. To implement the hierarchical structure of the HP ontology, one can use Information Content-based techniques, such as Resnik [15]. The information content (IC) of an HPO term $x$ is defined using the frequency of annotations related to that term within a given corpus of diseases (for instance OMIM or Orphanet),

$$IC(x) = -\log(p(x)), \tag{2}$$

where $p(x)$ is the frequency of $x$ in the diseases corpus (not to be confused with the frequency of a symptom, which we find in our database).

In this sense, it is a measure of the specificity of the symptoms: a specific symptom has a higher IC compared to a general symptom. Then, the similarity between two symptoms (i.e. two HPO terms) is calculated as the IC of their most informative common ancestor,

$$sim(x_1, x_2) = \max_{y \in C(x_1, x_2)} IC(y), \tag{3}$$

where $C(x_1, x_2)$ is the set of ancestor terms shared by symptoms $x_1$ and $x_2$. Finally, the similarity between two diseases $A$ and $B$ can be calculated as the average over the similarities between all the pairs $(x_i^A, x_j^B)$, where $x_i^A$ and $x_j^B$ are respectively symptoms of $A$ and $B$:

$$sim_{Resnik}(A, B) = avg[sim(x_i^A, x_j^B)]. \tag{4}$$

This combination of term similarities can be computed using several other averaging techniques. In this work, we used the best-match average (BMA), defined as

$$sim_{BMA}(A, B) = \frac{avg_{x_i^A}[\max_{x_j^B}(sim(x_i^A, x_j^B))] + avg_{x_j^B}[\max_{x_i^A}(sim(x_i^A, x_j^B))]}{2},$$
(5)

which prevents some diseases with a few high-IC associated terms from obtaining a high score in the rankings.

We also computed similarities using a graph-based approach called GraphIC [16], which again uses the information content of the symptoms. GraphIC computes the similarity between diseases $A$ and $B$ as an average of the information contents of the shared symptoms and all their ancestors. Let $A_{exp}$ and $B_{exp}$ be respectively the set of symptoms $A$ and $B$ expanded with all the ancestors of each HPO term in $A$ and $B$, up to the root node. Then, the GraphIC similarity between $A$ and $B$ is computed as

$$sim_{GraphIC}(A, B) = \frac{\sum_{x \in A_{exp} \cap B_{exp}} IC(x)}{\sum_{x \in A_{exp} \cup B_{exp}} IC(x)}.$$
(6)

Both the Resnik and GraphIC similarities have been computed using the PyHPO package (v4.0.0.) in Python.

Finally, we implemented another way to compare diseases based on topic modeling concepts and information retrieval (IR) techniques. Information retrieval techniques, commonly used in natural language processing and search engines, are designed to compare and rank documents based on their content. In this context, each disease was treated as a "document" and its associated HPO terms were interpreted as "words" describing its content. This approach allows to naturally implement the importance of each symptom using the frequency of each HPO term within the disease annotation dataset in the same way the occurrence of words is used in traditional IR. These frequencies can be transformed to further control the importance of each symptom; in this work, we used the term frequency–inverse document frequency (TF-IDF) weighting scheme [17], which emphasizes symptoms that are frequent within a specific disease but rare across the dataset. Using this representation, standard similarity measures such as cosine similarity were applied to compare Minamata disease profile with other diseases.

However, a limitation of standard Information Retrieval techniques is that they do not account for the hierarchical structure of the Human Phenotype Ontology (HPO). Similarly to Jaccard Index, they are not able to catch semantic similarity between terms; this means that if a disease does not share any specific HPO term with Minamata disease, the similarity score will be zero. To address this issue, we implemented query expansion, a technique commonly used in IR to enhance retrieval performance by adding semantically related terms to a query. In our context, the query corresponds to the set of HPO terms characterizing Minamata disease. We expanded this set by including all terms from the HPO ontology, assigning each an artificial frequency that decays with its distance from the closest original term in the Minamata set. Specifically, for any HPO term $y$, we defined its frequency $f_y$ as:

$$f_y = f_x \cdot \alpha^{d(x,y)},$$
(7)

where $f_x$ is the frequency of the closest Minamata disease symptom $x$, $d(x,y)$ is the shortest path length between terms $x$ and $y$ in the HPO graph, and $\alpha \in (0, 1)$ is a decay factor. We evaluated $\alpha \in \{0, 0.1, 0.3, 0.5, 0.7, 0.9, 1.0\}$ and computed the Kendall-$\tau$ correlation of the rankings obtained (S3 Fig) to assess the sensitivity of the results to this parameter. For $\alpha \approx 0.3$–$0.7$, the rankings are highly correlated ($\tau \geq 0.9$); for this reason, in this work we used $\alpha = 0.5$ as trade-off between specificity and inclusion of neighbouring phenotypes. For example, consider the HPO term *Lower limb hypertonia*, which is not present in the original Minamata set. It is connected to *Spasticity* (present in the original set with frequency $f_{x_1}$) by a path of length $d = 3$: *Spasticity → Hypertonia → Limb hypertonia → Lower limb hypertonia*. However, it also related to the term *Hypotonia* with path length $d = 4$ (by the path *Hypotonia → Abnormal muscle tone → Hypertonia → Limb hypertonia*

→ *Lower limb hypertonia*) and frequency $f_{x_2}$. Since *Spasticity* is the closest term, we include *Lower limb hypertonia* in the query with frequency $f_y = f_{x_1} \cdot 0.5^3$. Query expansion solves the problem of having many diseases with a zero similarity score, as one can clearly see in Fig 2a.

## Correlation of similarity measures

To understand how these different similarity measures relate to each other, we computed pairwise Kendall rank correlation coefficients. This coefficient is a nonparametric statistical method used to measure the ordinal association between two ranked lists. It quantifies how similar the orderings of elements are in the two lists by comparing the number of concordant and discordant pairs. A pair of elements is concordant if their order is the same in both rankings, and discordant if their order is reversed. The Kendall tau coefficient $\tau$ ranges from $-1$ (complete disagreement) to $+1$ (complete agreement), with 0 indicating no correlation. Since we are mainly interested in the diseases most similar to Minamata disease, we used a weighted version of Kendall Tau correlation (exchanges in the top spots are more influential than exchanges in the lower spots). In particular, we computed the correlation between rankings using the *weightedtau* function from scipy.stats with hyperbolic weighting function (i.e. rank $r$ is mapped to weight $\frac{1}{r+1}$).

## Ranking aggregation and disease classification

To combine the various classifications obtained into a single consensus ranking, we aggregated the individual rankings assuming equal importance for each method. Of the two rankings obtained using TF-IDF, we only kept the one where Query Expansion was applied, as it reduced the number of disease-pairs having zero similarity (see Results section). Finally, in order to obtain a broader understanding of the types of diseases most similar to Minamata disease, we analyzed the disease categories of the top 50 most similar diseases identified by our methods. For each disease, we retrieved its classification within the MONDO ontology by tracing all paths from the disease node up to the root and collecting the associated high-level disease categories. We used the 42 main disease classes defined in MONDO as reference categories and counted the number of times each category appeared across the top 100 diseases.

## Results

We first computed the similarity between Minamata disease and 12,622 diseases based on the phenotypic profile using five approaches: Jaccard Index, Resnik similarity, GraphIC, TF-IDF, and TF-IDF with Query Expansion. Beyond the diseases common to all rankings, each method highlights different neighbors (S1 Table). Jaccard finds disorders sharing a few overlapping HPO terms regardless of specificity or frequency—e.g., X-linked spasticity–intellectual disability–epilepsy syndrome, peroxisome biogenesis disorder 9B, isolated cerebellar hypoplasia/agenesis, illustrating how profiles with generic neurological signs can score well when overlap exists. TF-IDF prioritizes phenotypes with rarer, more informative terms in the overall profile; for this reason, we see the emergence of diseases such as psychogenic movement disorders, choreatic disease, Spasmus nutans, neuronal intranuclear inclusion disease. Adding query expansion (TF-IDF+QE) spreads weight to nearby motor-control terms in the HPO graph and therefore pulls in other conditions of the parkinsonian/dystonia spectra such as corticobasal syndrome, atypical juvenile parkinsonism, torsion dystonia 13, and dyskinesia with orofacial involvement. Resnik can be driven by single highly specific shared symptoms (via high-IC ancestors), which explains outliers like burning mouth syndrome. Ranking-specific Parkinsonian diseases are autosomal recessive early-onset Parkinson disease 6, parkinson disease 25 and autosomal recessive early-onset Parkinson disease 7. GraphIC, by averaging IC over shared terms and their ancestors, highlights disorders with broader overlap across neurological and metabolic ancestry such as Susac syndrome, Huntington disease–like 2, and GLUT1-deficiency encephalopathy. Together, these method-specific lists clarify how overlap-based, IR-based, and IC-based definitions of similarity emphasize different facets of Minamata's phenotype. Each technique emphasized different aspects of phenotypic similarity—overlap, ontological depth, or term specificity. The resulting phenotypic profile of Minamata disease is shown in Fig 1.

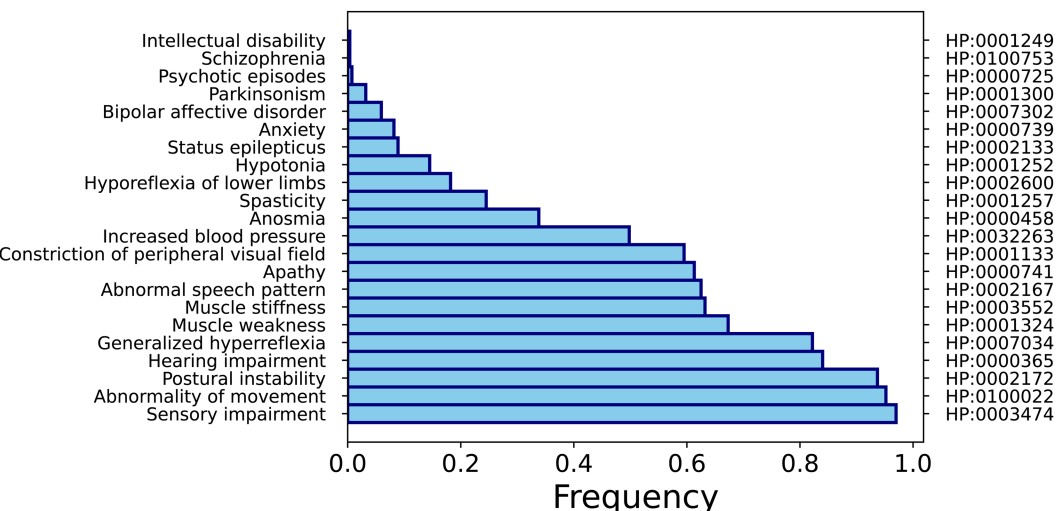

**Fig 1. Phenotypic profile of Minamata disease.** The symptoms of Minamata disease - as mapped from the original dataset to their corresponding HPO terms - and their frequency.

As shown in Fig 2a, similarity scores from all methods decrease sharply after the top 100 most similar diseases, indicating that only a small fraction of diseases are phenotypically close to Minamata disease. This steep decline suggests that the highest-ranking diseases are not only more similar in absolute terms but are also meaningfully distinct from the rest of the dataset. In other words, the top matches likely reflect genuine phenotypic overlap rather than noise or spurious similarities, reinforcing the biological relevance and interpretability of the similarity rankings. Jaccard and plain TF-IDF scores dropped to zero for a large portion of the dataset, due to their inability to handle semantically related but non-identical HPO terms. However, the decline behaves differently in each case: while TF-IDF scores decrease gradually, Jaccard

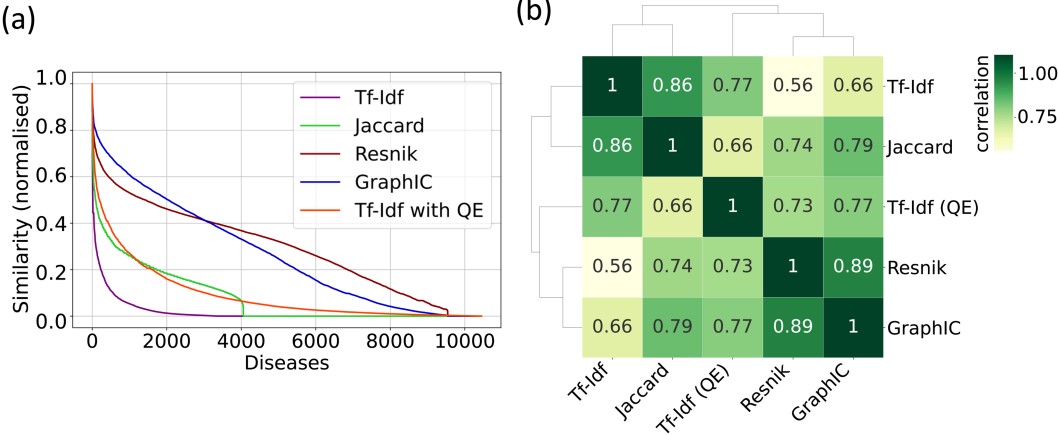

**Fig 2. Relationship between different rankings.** (a) Sorted scores of similarity with Minamata Disease, divided by their maximum value. The five curves are obtained with different techniques: cosine similarity combined with TF-IDF (orange line) and without (purple line) Query Expansion, Jaccard Index of shared symptoms (green line), Resnik (dark-red line), and GraphIC similarity (blue line). Lower panel shows the same curves with log-scale on the x-axis. (b) Weighted Kendall rank correlation coefficient between different similarity rankings. Darker squares represent stronger ranking correlation. Methods that yield more similar rankings appear closer together, as determined by hierarchical clustering based on pairwise correlation.

scores fall abruptly to zero once there is no direct symptom overlap between Minamata disease and the remaining diseases. For TF-IDF, this issue was mitigated using Query Expansion by including all the terms in the Human Phenotype Ontology with an artificial frequency that reflects the graph-wise distance from the original terms.

The top-scoring diseases varied between methods. Notably, several movement and neurodegenerative disorders, including *cyanide-induced parkinsonism*, *progressive supranuclear palsy*, and *early-onset parkinsonism–intellectual disability syndrome*, appeared consistently in the top 10 of multiple rankings. Moreover, five diseases appeared among the top 100 most similar diseases across all methods: *cyanide-induced parkinsonism*, *corticobasal syndrome*, *encephalopathy due to GLUT1 deficiency*, *progressive supranuclear palsy-parkinsonism syndrome* and *young-onset Parkinson disease*.

To compare how different similarity methods ranked diseases, we computed the weighted Kendall tau correlation between rankings (Fig 2b). Recall that a hyperbolic weighting function was used to give greater importance to differences occurring near the top of the rankings, since the top-ranked diseases are the most relevant for our analysis. All the correlations were above 0.5, with the lowest one being between the Resnik and TF-IDF rankings, likely due to the many zero-similarity entries in the latter. Conversely, the highest correlations (around 0.9) were observed between Resnik and GraphIC, both IC-based methods. It is worth noting that incorporating Query Expansion significantly improved TF-IDF's alignment with IC-based methods; for instance, its correlation with the GraphIC ranking is around 0.8. Surprisingly, the simplest technique used, namely the Jaccard Index, is quite correlated with all the other rankings, with a correlation that is never lower than 0.65.

To consolidate the results obtained from the various similarity measures, we applied a rank aggregation approach. We constructed a consensus ranking using weighted rank aggregation with equal weights across methods (Fig 3a); this allowed us to reduce the influence of method-specific biases and obtain a more balanced and comprehensive similarity assessment. The top disease in this final ranking was *cyanide-induced parkinsonism*, followed by several neurodegenerative and motor coordination disorders. It is worth mentioning that *cyanide-induced parkinsonism* figures in the top six most similar diseases of all rankings considered, with the only exception of GraphIC.

To better understand the types of diseases most similar to Minamata disease, we traced each of the top 50 diseases to their high-level MONDO categories. As shown in Fig 3b, the most represented categories were nervous system disorders, hereditary diseases, and movement disorders, followed by metabolic and psychiatric disorders. These results align with known clinical features of Minamata disease.

Conclusions were stable across unknown-frequency choices (0.01 and 1): in particular, top diseases and disease families, as well as the cross-method concordance, were preserved (S1 and S2 Figs).

## Discussion

The Minamata disease is a neurological severe disorder caused by the long-term exposure to high concentration of mercury. The exceptional conditions leading to Minamata disorder was used to highlight which pathological conditions with similar symptoms could be linked to heavy metal pollution, specifically mercury. One of the most complex issues for a doctor who has to make a diagnosis, is to understand the cause of a disease, especially when it is possibly related to environmental factors. Understanding of the cause of the symptoms in a patient can help the doctor clarify the nature of the disease and identify the best therapeutic solution. We, therefore, investigated possible associations of the Minamata's symptoms to the HBO database which includes over 13000 terms describing various abnormalities and features in human diseases.

Since the late 1990s, the development and adoption of standardized biomedical ontologies—such as the Gene Ontology [18], the Human Phenotype Ontology [7], MONDO [19], and others—have steadily grown, enabling systematic and computational approaches to disease and phenotype comparison. In many recent studies, standardized ontologies have been used to integrate diverse types of data—including phenotypic, genomic, molecular, and disease information—to

(a)

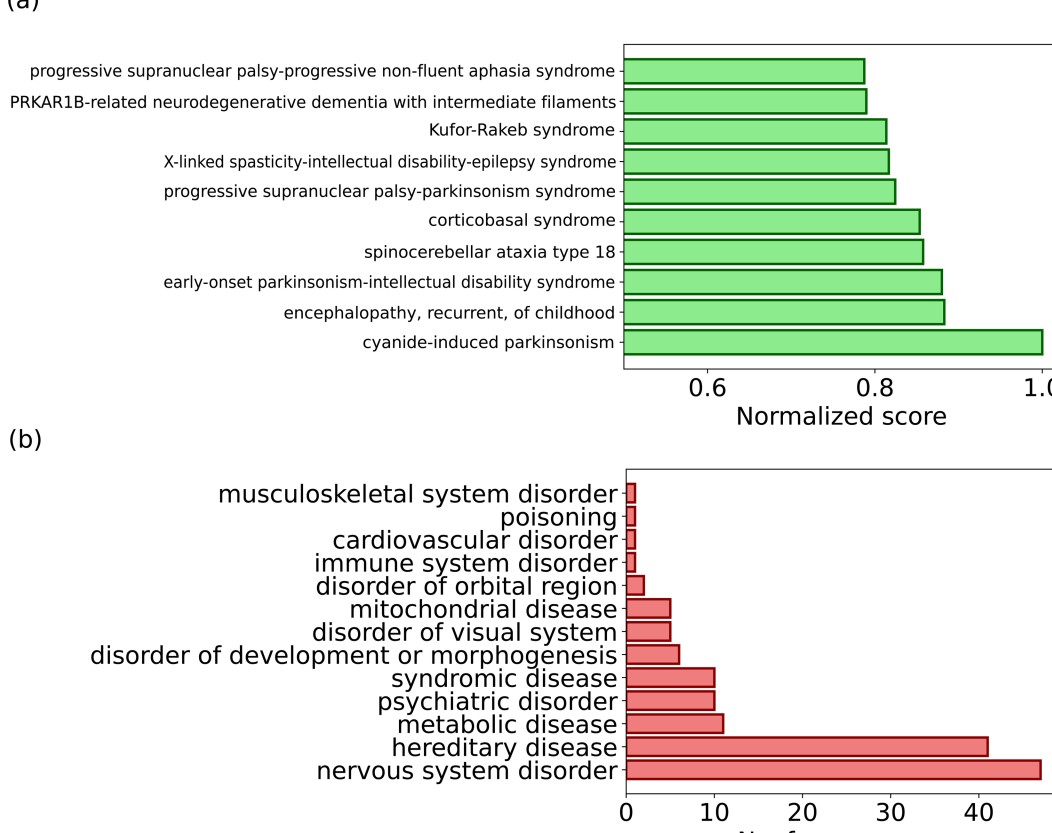

(b)

**Fig 3**. **Aggregate ranking and disease classification.** (a) The final ranking of diseases most similar to Minamata obtained by aggregating the rankings corresponding to Jaccard Index, Resnik, GraphIC and TF-IDF (with QE). The scores are normalised by the score of the most similar disease (*cyanide-induced parkinsonism*). (b) Histogram of the disease categories (according to the classification of the MONDO ontology) for the top 50 diseases most similar to Minamata Disease in the final ranking obtained by combining different rankings using Weighted Rank Aggregation.

develop powerful tools for diagnosis and disease classification, for instance to infer disease relationships or prioritize candidate genes [20–22]. The Minamata disorder was not included in databases such as OMIM or HPO and has unknown gene associations, molecular profiles, or large-scale imaging data. We, therefore, used exclusively phenotypic information, manually extracted from historical clinical reports and mapped into standardized ontology terms. By translating its clinical profile into standardized Human Phenotype Ontology (HPO) terms and comparing it to over 13,000 diseases using a range of similarity metrics, we explored how this environmentally induced condition relates to genetically defined disorders in terms of symptomatology.

Minamata's disorder was a unique case in history due to the high concentration of the mercury given the high concentration and long time exposure to which people were exposed. The symptoms were multiple and involved sensory system, movement and postural instability, hearing impairment but also weakness and stiffness of the muscles, abnormal speech, apathy, problems at the peripheral visual field and an increase of blood pressure. The comparison of Minamata symptoms with diseases in HPO with different methods including ontology-aware metrics such as Resnik similarity and GraphIC, and term-based information retrieval (IR) techniques, allowed us to identify a list of diseases that are close to Minamata's disorder. We found a strong similarity to cyanide-induced parkinsonism, encephalopathy, spinocerebellar ataxia type, neurodegenerative dementia related to PRKAR1B and aphasia syndrome progressive supranuclear palsy. The connection

was as expected related to nervous system disorder and less related to psychiatric or other diseases such as metabolic, cardiovascular or immune disorders.

Cyanide-induced parkinsonims is a rare parkinsonian syndrome due to intoxication which develops in individuals surviving an acute cyanide intoxication episode or due to chronic exposure to small cyanide doses [23]. The strong correlation of this rare pathology with Minamata's disorder shows how high concentration of mercury can produce the same symptoms. This information can be useful to doctors since it shows that these symptoms are not only linked to cyanide poisoning, but that mercury, if at high and prolonged doses, could also lead to similar effects. Interestingly, spinocerebellar ataxia type is a genetic progressive disease that comprises a group of neurodegenerative multisystem disorders, characterized by slowly progressive gait ataxia and variable additional symptoms including visual problems, Parkinsonism, dystonia, peripheral neuropathy [24]. Since there is no evidence of connection with mercury for this specific type of ataxia, this information might help clinicians in their diagnosis. Neurodegenerative dementia related to PRKAR1Ba is a rare, genetic neurodegenerative disease characterized by dementia and mild parkinsonism [25], without any clear connection with mercury. In this case, the connection with Minamata's disorder is intriguing but might be coincidental. Aphasia syndrome progressive supranuclear palsy is a rare atypical variant of Progressive Supranuclear Palsy (PSP) where the initial symptom is a speech and language disorder (apraxia of speech, agrammatism, and phonemic errors) [26]. Finally, encephalopathy is known to be related to mercury poisoning due to long term exposure to low doses of mercury [27].

A detailed analysis of S2 Table distinguishes between diseases linked by shared pathophysiological pathways and those sharing only superficial clinical features. Cyanide-induced parkinsonism stands out as a primary example of environmental neurotoxicology, scoring highly across multiple metrics because its toxin-mediated disruption of mitochondrial function creates a profound mechanistic overlap with genetic etiologies such as PINK1 and DJ-1 (Parkinson 6 and 7), which are rooted in mitochondrial maintenance and oxidative stress response. Lateral sclerosis further bridges these categories, as its similarity score reflects deep-seated neurodegenerative mechanisms, such as mitochondrial and lysosomal dysfunction, shared by both its sporadic environmental forms and familial genetic variants. In contrast, matches such as psychogenic movement disorders are likely to represent coincidental phenotypic similarity, with clinical symptoms mimicking the search criteria without sharing the underlying cellular pathology of neuronal death. Finally, the list is dominated by rare genetic disorders like Kufor-Rakeb syndrome and PRKAR1B-related dementia, highlighting how recurring themes of metabolic and cellular failure drive the highest similarity rankings.

## Conclusions

In this work, we proposed a computational re-evaluation of Minamata disease clinical data suggesting a consistent phenotypic overlap with several established movement and neurodegenerative disorders. Through the application of multiple metrics, including Jaccard Index, Resnik, GraphIC, and TF-IDF with query expansion, our analysis identified a robust consensus of similar pathologies such as Cyanide-induced parkinsonism and progressive supranuclear palsy, corticobasal syndrome, and encephalopathy. These patterns of similarity suggest that the clinical symptoms observed in Minamata patients—such as postural instability, muscle stiffness, ataxia, and abnormal speech—may arise from the involvement of shared neuroanatomical systems, specifically the cerebellar and extrapyramidal pathways. This phenotypic convergence is interpreted as an indication of common symptomatic pathways rather than evidence of shared molecular or genetic mechanisms. For instance, while spinocerebellar ataxia shows high similarity, there is no established evidence linking mercury to its specific genetic causes; the association instead highlights clinical parallels.

It is necessary to acknowledge the inherent limitations of using phenotypic similarity as a standalone analytical tool. These rankings are intended to be hypothesis-generating and do not constitute definitive mechanistic proof. The study relied exclusively on phenotypic data because Minamata disease currently lacks the documented gene associations or molecular profiles available for the other diseases in the Human Phenotype Ontology. Therefore, the associations identified in this study require independent validation through future experimental and epidemiological investigations. Despite

these constraints, the workflow demonstrates how structured ontologies can contextualize legacy environmental disorders within a modern pathological framework. All together, our work suggests that re-examining legacy diseases through structured phenotype ontologies can yield meaningful insights and may prove valuable in studying other conditions with environmental, toxicological, or multifactorial origins. It also highlights the potential of information retrieval techniques in this context and offers a template for similar studies that must rely exclusively on phenotypic data.

## Supporting information

**S1 Table. HPO mapping of Minamata disease symptoms.** Formatted table of symptoms of patients from the Minamata Area diagnosed with Minamata disease as found in [13], with the associated frequency and the most closely related HPO terms. Notice that the frequency of "Mental retardation" was inconsistently annotated in [13]. To resolve the contradiction, we selcted the smaller reported frequency (0.37%).
(PDF)

**S2 Table. Similarity rankings.** Rankings of diseases most similar to Minamata disease using five approaches (Jaccard Index, TF-IDF, TF-IDF with query expansion, Resnik and GraphIC) with the associated scores.
(PDF)

**S1 Fig. Relationship between different rankings for $f = 0.1$.** (a) Sorted scores of similarity with Minamata Disease, divided by their maximum value. The five curves are obtained with different techniques: cosine similarity combined with TF-IDF (orange line) and without (purple line) Query Expansion, Jaccard Index of shared symptoms (green line), Resnik (dark-red line), and GraphIC similarity (blue line). Lower panel shows the same curves with log-scale on the x-axis. (b) Weighted Kendall rank correlation coefficient between different similarity rankings. (c) The final ranking of diseases most similar to Minamata obtained by aggregating the rankings corresponding to Jaccard Index, Resnik, GraphIC and TF-IDF (with QE). The scores are normalised by the score of the most similar disease (*cyanide-induced parkinsonism*). (d) Histogram of the disease categories (according to the classification of the MONDO ontology) for the top 50 diseases most similar to Minamata Disease in the final ranking obtained by combining different rankings using Weighted Rank Aggregation.
(TIFF)

**S2 Fig. Relationship between different rankings for $f = 1$.** (a) Sorted scores of similarity with Minamata Disease, divided by their maximum value. The five curves are obtained with different techniques: cosine similarity combined with TF-IDF (orange line) and without (purple line) Query Expansion, Jaccard Index of shared symptoms (green line), Resnik (dark-red line), and GraphIC similarity (blue line). Lower panel shows the same curves with log-scale on the x-axis. (b) Weighted Kendall rank correlation coefficient between different similarity rankings. (c) The final ranking of diseases most similar to Minamata obtained by aggregating the rankings corresponding to Jaccard Index, Resnik, GraphIC and TF-IDF (with QE). The scores are normalised by the score of the most similar disease (*cyanide-induced parkinsonism*). (d) Histogram of the disease categories (according to the classification of the MONDO ontology) for the top 50 diseases most similar to Minamata Disease in the final ranking obtained by combining different rankings using Weighted Rank Aggregation.
(TIFF)

**S3 Fig. Effect of the decay factor $\alpha$.** Wheighted Kendall rank coefficient of TF-TDF (with Query Expansion) rankings obtained with different values of the decay factor $\alpha$.
(TIFF)

## Author contributions

**Conceptualization:** Paolo Boldi, Elena Casiraghi, Stefano Zapperi, Caterina A. M. La Porta.

**Formal analysis:** Edoardo Marchi.

**Supervision:** Paolo Boldi, Elena Casiraghi, Stefano Zapperi, Caterina A. M. La Porta.

**Writing – original draft:** Caterina A. M. La Porta.

**Writing – review & editing:** Stefano Zapperi, Caterina A. M. La Porta.

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
