## [Editor Report · Decision Letter 0]

23 Jul 2025

PONE-D-25-32685Revisiting Minamata disease through computational phenotype similarity analysisPLOS ONE

Dear Dr. La Porta,

Thank you for submitting your manuscript to PLOS ONE. After careful consideration, we feel that it has merit but does not fully meet PLOS ONE’s publication criteria as it currently stands. Therefore, we invite you to submit a revised version of the manuscript that addresses the points raised during the review process.

In particular we would like you to include higher quality images as, at the moment, they are not of sufficient quality to be readable during the review process.

We look forward to receiving your revised manuscript.

Kind regards,

Nikolas Pontikos, PhD

Academic Editor

PLOS ONE

[CAMLP and EC acknowledge funding from FAIR - Future Artificial Intelligence Research: Adaptive AI methods for Digital Health" grant number PNRR\_BAC24GVALE\_01 PE\_0000013, CUP D53C2200238000].

Additional Editor Comments:

Please resubmit with higher quality images to facilitate the review process for our referees. It is currently not possible to read some of the figures.
---

## [Author Response · Author response to Decision Letter 1]

31 Jul 2025

see the rebuttal letter enclosed in the submssion

---

## [Decision Letter · Decision Letter 1]

1 Dec 2025

PONE-D-25-32685R1Revisiting Minamata disease through computational phenotypic similarity analysisPLOS ONE

Dear Dr. La Porta,

Thank you for submitting your manuscript to PLOS ONE. After careful consideration, we feel that it has merit but does not fully meet PLOS ONE’s publication criteria as it currently stands. Therefore, we invite you to submit a revised version of the manuscript that addresses the points raised during the review process.

We look forward to receiving your revised manuscript.

Kind regards,

Ejaz Ahmad Khan, M.D, MPH, FFPH

Academic Editor

PLOS ONE

Journal Requirements:

Reviewers' comments:

Reviewer's Responses to Questions

**Comments to the Author**

1. If the authors have adequately addressed your comments raised in a previous round of review and you feel that this manuscript is now acceptable for publication, you may indicate that here to bypass the “Comments to the Author” section, enter your conflict of interest statement in the “Confidential to Editor” section, and submit your "Accept" recommendation.

Reviewer #1: All comments have been addressed

Reviewer #2: (No Response)

2. Is the manuscript technically sound, and do the data support the conclusions?

Reviewer #1: Yes

Reviewer #2: Yes

3. Has the statistical analysis been performed appropriately and rigorously?

Reviewer #1: N/A

Reviewer #2: Yes

4. Have the authors made all data underlying the findings in their manuscript fully available?

Reviewer #1: Yes

Reviewer #2: Yes

5. Is the manuscript presented in an intelligible fashion and written in standard English?

Reviewer #1: Yes

Reviewer #2: Yes

6. Review Comments to the Author

Reviewer #1: There are many studies that use the similarity of HPO to evaluate disease distance. for example, RDmap: A Map for Exploring Rare Diseases. Orphanet Journal of Rare Diseases. 2021, 16(101). It calculates distances for thousands of rare diseases and visualizes them. It seems that this tool can be used to complete the search for diseases with similar phenotypes to a certain disease in this study.

The search for diseases with similar phenotypes is quite straightforward for rare diseases. The Minamata disease targeted in this study has nothing special about it. Clearly, if another disease were to be found, a list of similar diseases would also be obtained. These lists may imply some disease mechanisms, but all of these need to be verified through subsequent research. This research clearly lacks such a conclusion.

In addition, the core focus of the article is somewhat confusing. In the method section, a large number of similarity calculation methods are introduced, with the focus on the technical aspect. However, the results section does not pay much attention to these methods but instead obtains a comprehensive ranking list, which mainly presents various possible hypotheses. Essentially, this research lacks a conclusion.

Reviewer #2: The manuscript presents a clear and well‐structured computational analysis that maps historical Minamata disease symptoms to HPO terms and compares them against >12,000 diseases using multiple similarity metrics. The scope and framing align with PLOS ONE’s focus on technical rigor and ethical standards rather than perceived impact.

The methods are described in commendable detail, including ontology mapping, frequency harmonization, and rank aggregation. Two preprocessing choices would benefit from small additions to improve reproducibility:

1. Provide a brief sensitivity analysis showing the effect of assigning frequency 0.5 to symptoms lacking data (currently justified as an uninformative prior)

2. Justify the query‐expansion decay factor α = 0.5 with either literature precedent or a data‐driven comparison (e.g., α ∈ {0.3, 0.5, 0.7}) on rank stability.

The results are compelling and consistently converge on movement and neurodegenerative disorders, with cyanide‐induced parkinsonism repeatedly appearing near the top of multiple rankings. Figure 2b’s weighted Kendall τ visualization effectively communicates agreement among metrics, and Figure 3’s consensus ranking is appropriate. To strengthen interpretation while remaining within PLOS ONE’s rigor criteria, consider adding a short paragraph that distinguishes likely mechanistic overlap (for example, mitochondrial dysfunction, oxidative stress) from coincidental phenotypic similarity, and that clarifies which top matches plausibly relate to environmental neurotoxicology versus genetic etiologies. This will better align claims to the supporting data and help readers understand the limits of phenotypic similarity alone. �

Finally, a few presentation details will improve clarity without altering conclusions. In the "Minamata Disease symptoms dataset" please cite the specific tables used to resolve the discrepancy for "Mental Retardation" and, if possible, add a short note in S1 Table that flags this correction. In the figures, ensure axis labels are fully legible in the final PDF.

7. PLOS authors have the option to publish the peer review history of their article (what does this mean?). If published, this will include your full peer review and any attached files.

Reviewer #1: No

Reviewer #2: No

---

## [Author Response · Author response to Decision Letter 2]

5 Jan 2026

Response to Reviewer #1

Concern 1: There are many studies that use the similarity of HPO to evaluate disease distance. for example, RDmap: A Map for Exploring Rare Diseases. Orphanet Journal of Rare Diseases. 2021, 16(101). It calculates distances for thousands of rare diseases and visualizes them. It seems that this tool can be used to complete the search for diseases with similar phenotypes to a certain disease in this study.

Answer: We thank the reviewer for pointing us to RDmap and related work. We now explicitly cite RDmap in the Introduction and discuss how our study relates to this line of research. While tools such as RDmap are invaluable for the interactive exploration of rare diseases already present in ontological databases, our work is unique in its ability to integrate and leverage unstructured historical clinical data. Our methodology can be extended to other environmental or historical diseases not yet formalized in major databases, providing a replicable approach for extracting value from legacy epidemiological data. In particular, our work focuses on Minamata disease as a case study of a disorder that is not natively represented in HPO or OMIM and whose clinical description originates from historical epidemiological surveys.

We have added a paragraph in the introduction clarifying this distinction and framing our study as complementary to RDmap—relying on similar ontological infrastructure but addressing a different problem setting and data type. See page 2 lines 39-42.

Concern 2: The search for diseases with similar phenotypes is quite straightforward for rare diseases. The Minamata disease targeted in this study has nothing special about it. Clearly, if another disease were to be found, a list of similar diseases would also be obtained. These lists may imply some disease mechanisms, but all of these need to be verified through subsequent research. This research clearly lacks such a conclusion.

Answer: We agree that, from a technical standpoint, once diseases are represented in a standardized ontology such as HPO, searching for phenotypically similar disorders is conceptually straightforward and can be applied to any index disease. The added value of our work lies in the practical demonstration of how historical epidemiological data can be integrated into the context of modern ontologies, enabling systematic comparisons and the generation of new hypotheses. Specifically, the application to Minamata disease shows how it is possible to explore the position of an environmental disease within the landscape of neurological disorders, even in the absence of genetic or molecular data. This approach may be useful for other environmental or multifactorial diseases, which are often overlooked by genetic databases.

In the revised version of the manuscript, we have better clarified in the Introduction and Discussion that:

• Our goal is not to introduce a novel similarity algorithm, but to demonstrate a reproducible workflow for translating historical clinical descriptions (here, Minamata disease) into ontological form and contextualizing them within the modern landscape of >12,000 HPO-annotated diseases.

• Minamata disease serves as an important case study because it represents a prototypical environmental neurotoxic disorder with a well-documented clinical picture, yet it is absent from modern disease ontologies and databases.

• The resulting similarity rankings are intended as hypothesis-generating, not as definitive mechanistic proofs. We now state explicitly that any mechanistic inferences suggested by phenotypic similarity must be tested in future experimental and epidemiological studies.

We discuss the point above at end of the Introduction page 2 lines 45-52.

Furthermore, to address the concern about the lack of clear conclusions, we have added a short Conclusions section (page 10) that:

• Summarizes the main consistent findings, including the convergence of multiple metrics on movement disorders and neurodegenerative or neurotoxic syndromes (notably cyanide-induced parkinsonism).

• Emphasizes that these patterns support the plausibility of shared affected neuroanatomical systems (e.g., cerebellar and extrapyramidal involvement) without asserting a shared molecular mechanism.

• Clearly delineates the limitations of phenotypic similarity as a standalone tool and the need for independent validation.

Concern 3: In addition, the core focus of the article is somewhat confusing. In the method section, a large number of similarity calculation methods are introduced, with the focus on the technical aspect. However, the results section does not pay much attention to these methods but instead obtains a comprehensive ranking list, which mainly presents various possible hypotheses. Essentially, this research lacks a conclusion.

Answer: We appreciate this comment and have revised parts of the manuscript to clarify the core focus and improve the alignment between Methods and Results. In the revised Introduction, we now explicitly state that the main objective is to derive robust disease rankings by combining multiple complementary similarity metrics and then interpret the resulting consensus in terms of disease categories and neurotoxic vs genetic etiologies (page 2). This makes clear why we devoted less space to different metrics, but ultimately centered the results on consensus patterns. In the revised Results section (pages 6-7 lines 225-245), we now make more explicit reference to how the individual metrics behave and where they agree or differ, rather than immediately focusing only on the aggregated ranking. As mentioned above, we added a more explicit Conclusions section that synthesizes the methodological and biological insights and clarifies the scope and limits of the work.

Reviewer #2: The manuscript presents a clear and well‐structured computational analysis that maps historical Minamata disease symptoms to HPO terms and compares them against >12,000 diseases using multiple similarity metrics. The scope and framing align with PLOS ONE's focus on technical rigor and ethical standards rather than perceived impact.

The methods are described in commendable detail, including ontology mapping, frequency harmonization, and rank aggregation.

Concern 1: Two preprocessing choices would benefit from small additions to improve reproducibility:

1. Provide a brief sensitivity analysis showing the effect of assigning frequency 0.5 to symptoms lacking data (currently justified as an uninformative prior)

Answer: We now report a sensitivity analysis using two extreme values that correspond to HPO frequency classes: 0.01 (“very rare”, HP:0040284) and 1.0 (“obligate”, HP:0040280). For each choice, we recomputed all similarity rankings and the main figures. Conclusions are qualitatively unchanged: (i) the sharp score drop after ~100 diseases persists; (ii) cross-method rank concordance remains high and the pattern is the same of f=0.5; (iii) the same diseases and disease families remain at the top, although with slightly different rankings. We summarize these results in Fig. S1 and S2. We also add a short note in Results stating that our findings are robust even to these extreme bounds on unknown frequencies and a brief comment in Methods stating that a sensitivity analysis has been performed (page 3 lines 83-85).

2. Justify the query‐expansion decay factor α = 0.5 with either literature precedent or a data‐driven comparison (e.g., α ∈ {0.3, 0.5, 0.7}) on rank stability.

Answer: The factor α acts as a trade-off between specificity (low α) and inclusion of neighbouring phenotypes (high α). We performed a comparison over α ∈ {0 (no QE), 0.1, 0.3, 0.5, 0.7, 0.9, 1.0}. The Kendall-τ heatmap (included as Fig. S3) shows a stability plateau for α≈0.3–0.7. Rankings at α=0.5 are highly concordant with α=0.3 (τ≈0.96) and α=0.7 (τ≈0.94), and remain well aligned with the non-expanded TF-IDF baseline (τ≈0.77) which corresponds to α=0. Concordance drops as α→1.0, which corresponds to the case where all neighbours are assigned the same frequency of the original phenotypes. We therefore keep α = 0.5 as a neutral mid-range value within this plateau, representing a fair trade-off between preserving specificity and capturing nearby ontology neighbours. We now justify this choice in the Methods section (page 5 lines 183-186). The results are compelling and consistently converge on movement and neurodegenerative disorders, with cyanide‐induced parkinsonism repeatedly appearing near the top of multiple rankings. Figure 2b's weighted Kendall τ visualization effectively communicates agreement among metrics, and Figure 3's consensus ranking is appropriate.

Concern 2: To strengthen interpretation while remaining within PLOS ONE's rigor criteria, consider adding a short paragraph that distinguishes likely mechanistic overlap (for example, mitochondrial dysfunction, oxidative stress) from coincidental phenotypic similarity, and that clarifies which top matches plausibly relate to environmental neurotoxicology versus genetic etiologies. This will better align claims to the supporting data and help readers understand the limits of phenotypic similarity alone. �

Answer: We added a brief interpretive paragraph to Discussion distinguishing:

(i) toxin-linked entities (e.g., cyanide-induced parkinsonism; mercury-related encephalopathy), which are compatible with environmental neurotoxicology mechanisms, and

(ii) genetic neurodegenerative disorders (e.g., PRKAR1B-related dementia...), where proximity likely coincidental phenotypic similarity rather than shared etiology.

See pages 9-10 lines 359-374.

Concern 3:

Finally, a few presentation details will improve clarity without altering conclusions. In the "Minamata Disease symptoms dataset" please cite the specific tables used to resolve the discrepancy for "Mental Retardation" and, if possible, add a short note in S1 Table that flags this correction. In the figures, ensure axis labels are fully legible in the final PDF.

Answer: In Minamata Disease symptoms dataset, we now cite the exact tables used to resolve the “Mental Retardation” frequency discrepancy and add a note in the caption of Table S1 flagging the corrected frequency. See page 4 lines 111-115 and caption of Table S1

---

## [Decision Letter · Decision Letter 2]

28 Jan 2026

Revisiting Minamata disease through computational phenotypic similarity analysis

PONE-D-25-32685R2

Dear Dr. La Porta,

We’re pleased to inform you that your manuscript has been judged scientifically suitable for publication and will be formally accepted for publication once it meets all outstanding technical requirements.

Within one week, you’ll receive an e-mail detailing the required amendments. When these have been addressed, you’ll receive a formal acceptance letter, and your manuscript will be scheduled for publication.

An invoice will be generated when your article is formally accepted. Please note, if your institution has a publishing partnership with PLOS and your article meets the relevant criteria, all or part of your publication costs will be covered. Please make sure your user information is up to date by logging into Editorial Manager at Editorial Manager® and clicking the ‘Update My Information' link at the top of the page. For questions related to billing, please contact billing support.

Kind regards,

Ejaz Ahmad Khan, M.D, MPH, FFPH

Academic Editor

PLOS One

Additional Editor Comments (optional):

Reviewers' comments:

Reviewer's Responses to Questions

**Comments to the Author**

1. If the authors have adequately addressed your comments raised in a previous round of review and you feel that this manuscript is now acceptable for publication, you may indicate that here to bypass the “Comments to the Author” section, enter your conflict of interest statement in the “Confidential to Editor” section, and submit your "Accept" recommendation.

Reviewer #1: All comments have been addressed

Reviewer #2: All comments have been addressed

2. Is the manuscript technically sound, and do the data support the conclusions?

Reviewer #1: Yes

Reviewer #2: Yes

3. Has the statistical analysis been performed appropriately and rigorously?

Reviewer #1: Yes

Reviewer #2: Yes

4. Have the authors made all data underlying the findings in their manuscript fully available?

Reviewer #1: Yes

Reviewer #2: Yes

5. Is the manuscript presented in an intelligible fashion and written in standard English?

Reviewer #1: Yes

Reviewer #2: Yes

6. Review Comments to the Author

Reviewer #1: Thank you, author, for your responses to all the concerns and questions. I have no further questions.

Reviewer #2: The authors have addressed all comments raised in the previous review round.

In particular, the additional sensitivity analyses on symptom frequency assignment and the justification of the query expansion decay factor substantially improve the methodological transparency and reproducibility of the study. The new supplementary figures convincingly demonstrate that the main conclusions are robust to extreme parameter choices.

The revised Introduction and Discussion now more clearly articulate the scope of the work, explicitly framing the similarity rankings as hypothesis-generating rather than mechanistic proof. The added interpretive paragraph distinguishing environmentally driven neurotoxic conditions from genetically determined disorders effectively aligns the biological interpretation with the limits of phenotypic similarity analysis.

Minor presentation issues noted previously have also been addressed, including explicit citation of source tables for the "Mental Retardation" frequency discrepancy and improved clarity in figure labeling.

7. PLOS authors have the option to publish the peer review history of their article (what does this mean?). If published, this will include your full peer review and any attached files.

Reviewer #1: **Yes:** Haomin Li

Reviewer #2: No

---

## [Editor Report · Acceptance letter]

PONE-D-25-32685R2

PLOS One

Dear Dr. La Porta,

I'm pleased to inform you that your manuscript has been deemed suitable for publication in PLOS One. Congratulations! Your manuscript is now being handed over to our production team.

Kind regards,

on behalf of

Dr. Ejaz Ahmad Khan

Academic Editor

PLOS One